# MET is a new confirmed gene responsible for familial distal arthrogryposis

Debora Maffeo [1,2], Anna Carrer [1,2,3], Angela Rina [1,2], Loredaria Adamo [1,2], Caterina Lo Rizzo[3], Mirella Bruttini [1,2,3], Alessandra Renieri[1,2,3] & Francesca Mari [1,2,3 ✉]

Arthrogryposis is a congenital joint contracture involving more than one body area with an overall incidence of 1 in 3000. It can be isolated or in the context of syndromic conditions, with different grades of severity (Pehlivan et al, 2019). It is rather a sign than a diagnosis. It arises due to limited fetal movements because of either an intrinsic fetal neurological or muscular defect or extrinsic factors (Hall JG et al, 2019). The condition is highly genetic heterogeneous with more than 400 genes identified so far (Pehlivan et al, 2019). However, the molecular etiology remains unclear in many cases (Pehlivan et al, 2019).

In 2019, a study reported a single four-generation Chinese family with arthrogryposis with only upper limb involvement and a missense mutation of the *MET* gene affecting the Tyrosine residue 1234 (p.Tyr1234Cys) (Zhou et al, 2019). The authors generated a Met p.Y1232C (corresponding to the p.Y1234C in humans) transgenic mouse model identifying a defective migration of muscle progenitor cells and an impaired proliferation of secondary myoblasts. These data allowed to provisionally relate the *MET* gene to arthrogryposis (OMIM, 620019, Phenotype ?Arthrogryposis, distal, type 11).

MET (mesenchymal–epithelial transition factor) is a well-known proto-oncogene with somatic activating variants reported in several different cancers (Guo et al, 2020). Under normal conditions, it has been demonstrated to be crucial for embryogenesis, tissue regeneration, wound healing, and the formation of nerves and muscles (Maina et al, 1996; Haines et al, 2004). c-MET belongs to the receptor tyrosine kinase family. It is produced as a single-chain precursor which is proteolytically cleaved into a highly glycosylated extracellular α-subunit and a transmembrane β-subunit (Fig. 1C). The α-subunit contains a Sema domain which is linked to the Sema domain of the β-subunit by a disulfide bridge (Fig. 1C). The β-subunit is also composed of a PSI domain, and four IPT domains in the extracellular region, a transmembrane helix and an intracellular portion which contains a juxtamembrane domain, a kinase domain and a C-terminal tail with a multifunctional docking site (Fig. 1C). The kinase domain contains the catalytic tyrosines 1234 and 1235, which positively modulate enzyme activity, while the juxtamembrane domain contains tyrosine 1003, which negatively regulates c-MET (Fig. 1C). The C-terminal multifunctional docking site contains tyrosines Y1349 and Y1356, which recruit several transducers and adaptors when c-MET is active (Fig. 1C). Upon the hepatocyte growth factor (HGF) binding, the only known cMET ligand, the receptor is activated through the auto-phosphorylation of Tyr-1234 and Tyr-1235, leading to the phosphorylation of Tyr-1349 and Tyr-1356. This facilitates the recruitment of intracellular effector molecules and consequently the activation of downstream signaling pathways (Zhang et al, 2018).

Here, we report a second two-generation family with distal arthrogryposis and a *MET* mutation (Fig. 1A). The proband, an 18-year-old boy, presented bilateral hand arthrogryposis. He presented with diffuse arthrogryposis in both hands with inability to completely extend and bend fingers, absent flexion creases bilaterally, bilateral thenar and hypothenar hypotrophy, bilateral muscular hypotrophy of the distal two-thirds of the forearm and limited forearm supination (Fig. 1B,IV:3). Specifically, in both hands, he presented inability to oppose the thumb, clinodactyly of the II and V ray, flexion contraction of all interphalangeal joints of II–III–IV and V rays. In the left hand, he presented a scar on the ulnar side due to a previous orthopedic surgery. On the lower limb he presented bilateral pes cavum and hallux valgus (Fig. 1B,IV:3). His mother presented a more severe form, also involving feet, with overriding fingers (Fig. 1B,III:4). She presented flexion contraction of all digits of both hands, bilateral thenar and hypothenar hypotrophy and distal forearm muscular hypotrophy (more severe on the right side) (Fig. 1B,III:4). In the right hand, the most severely affected, she reported a previous orthopedic surgery between the I and II ray to improve grip, she presented camptodactyly of the II ray due to a pterygium, she reported a previous orthopedic surgery of the V ray to possibly remove a pterygium that was totally limiting digit extension. On the feet, she showed the II finger overlapping the III bilaterally already present at birth, bilateral pes cavum. She reported a previous surgery for a III mallet toe on the left foot. MRIs have not been performed in any of the two affected individuals.

Clinical exome analysis was performed in the proband and his mother and identified the *MET* mutation NM_000245:c.3704 A > G;p.(Tyr1235Cys), in the heterozygous state, in both. Exome sequences were enriched with the Truesight one expanded Kit (Illumina). Sequences were generated on a NovaSeq 6000 (Illumina). Data analysis was performed using Illumina BaseSpace Basic software, followed by bioinformatic analysis with enGenome – eVai (CE-IVD) software. Mean target coverage depth was 215 (proband) and 232 (mother), respectively. The quality score of variant call was 49,64 (proband) and 49,42 (mother). Coverage (Alternate/Reference) was

[1]Medical Genetics, University of Siena, 53100 Siena, Italy. [2]Med Biotech Hub and Competence Center, Department of Medical Biotechnologies, University of Siena, 53100 Siena, Italy. [3]Genetica Medica, Azienda Ospedaliera universitaria Senese, 53100 Siena, Italy. ✉E-mail: francesca.mari@unisi.it
https://doi.org/10.1038/s44321-024-00044-y | Published online: 1 March 2024

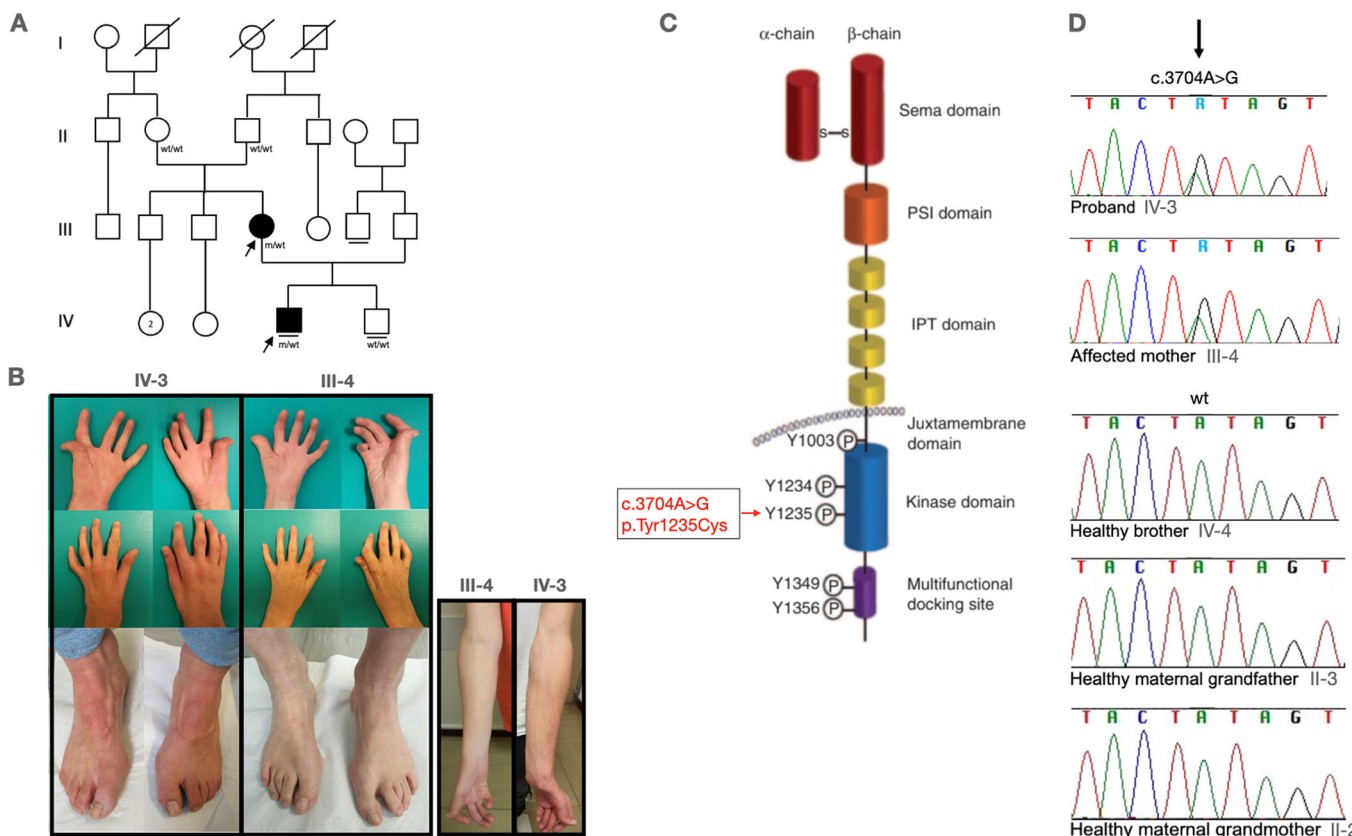

**Figure 1. *MET* p.Tyr1235Cys mutation caused distal arthrogryposis in a two-generation family.**

(A) Pedigree of the family. Filled symbols denote affected individuals, open symbols indicate unaffected individuals and symbols with slashes represent deceased individuals. Generations are indicated by roman numbers. The arrows indicate the proband (IV-3) and his affected mother (III-4). wt/wt = wild type genotype. m/wt = *MET* mutation in heterozygous state. (B) Pictures of affected individuals. Hands (dorsal and palmar view) and feet of the proband (left panel, IV:3) and his mother (panel, III:4). On the right, picture of forearm of affected individuals (IV-3 and III-4), showing hypotrophy. (C) Protein structure of c-MET. Sema = Semaphorin; PSI = plexin-semaphorin-integrin, and IPT = immunoglobulin-plexin-transcription factors. The arrow indicates the mutation site (Tyrosine residue 1235) identified in the family (image modified from Organ et al, 2011). (D) Sanger sequencing chromatograms of the mutated site (indicated by the arrow) in proband, affected mother, healthy brother and healthy grandparents showing the presence of the mutation (A > G substitution) in the affected subjects (IV-3 and III-4) and the wild-type sequence in the healthy subjects (IV-4, II-3, II-2).

124 (58/66) (proband) and 150 (69/81) (mother). No additional variants in genes associated with arthrogryposis were identified. Sanger sequencing confirmed the mutation in the two affected subjects and in none of the tested healthy family members (brother and maternal grandparents), indicating the possible de novo occurrence of the mutation in the proband's mother (Fig. 1D).

The affected residue is adjacent to the tyrosine mutated in the previously reported Chinese family (Zhou et al, 2019). Functional studies on *MET* p.Y1234 mutation showed the inability of the mutant MET receptor to phosphorylate residues Y-1234/1235, Y-1349, and Y-1356 in response to HGF treatment, resulting in failure of c-MET activation and decreasing of its kinase activity, with the consequent impairment of the HGF-MET signaling (Zhou

et al, 2019). A similar pathogenic mechanism is then anticipated for the mutation identified in our family.

Further studies are needed to investigate if these two residues are the only sites responsible for arthrogryposis or if mutations in the other functional tyrosines of the gene could determine the same phenotype. It would also be interesting to investigate whether phosphorylation impairment is the only pathogenic mechanism that determines *MET*-related arthrogryposis or whether it is the consequent alteration of the HGF-MET signaling pathway that is responsible for determining the phenotype. In this case, other types of variants in the same gene or in other genes of the same pathway that have the effect of disrupting HGF-MET signaling could also lead to the same phenotype.

From a clinical point of view, the two families show a consistent phenotype.

However, in the proband's mother, the lower limbs were also involved. This observation may broaden the phenotype associated with mutations in the *MET* gene and anticipate the clinical heterogeneity of *MET*-related arthrogryposis. However, given that in our family the characteristics of the feet in the two affected subjects are not completely consistent, we cannot exclude that lower limbs abnormalities could be due to a different cause and not be part of the *MET*-related phenotype. Additional reports will help define this.

In conclusion, our results provide the needed evidence to define *MET* as a new gene causative for isolated distal arthrogryposis.

## Study approval

Informed consent for photos and data publication was obtained from all subjects.

The experiments were conformed to the principles set out in the WMA Declaration of Helsinki and the Department of Health and Human Services Belmont Report.

## Data availability

The authors declare that the data supporting the findings of this study are available within the paper. Exome data for individual patients cannot be made publicly available for reasons of patient confidentiality. Qualified researchers may apply for access to these data, pending institutional review board approval.

## Peer review information

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

## Acknowledgements

The authors would like to express our gratitude to the patients who participated in this study. This work was in part supported by EU funding within the NextGenerationEU-MUR PNRR Tuscany Health Ecosystem (Project no ECS00000017-THE) to FM. The specimens were provided by the Genetic Biobank of Siena, member of Telethon Network of Genetic Biobanks (TNGB, project no. GTB18001), EuroBioBank, and RD-Connect.

## Author contributions

**Debora Maffeo**: Data curation; Formal analysis; Methodology; Writing—original draft. **Anna Carrer**: Data curation; Investigation. **Angela Rina**: Formal analysis. **Loredaria Adamo**: Formal analysis; Methodology. **Caterina Lo Rizzo**: Formal analysis; Investigation. **Mirella Bruttini**: Formal analysis; Supervision; Validation. **Alessandra Renieri**: Supervision; Validation. **Francesca Mari**: Conceptualization; Data curation; Supervision; Funding acquisition; Validation; Writing—original draft; Writing—review and editing.

## Disclosure and competing interests statement

The authors declare no competing interests.

