## [Peer Review File · EMBO Molecular Medicine]

MET is a new confirmed gene responsible for familial distal arthrogryposis.

Debora Maffeo, Anna Carrer, Angela Rina, Loredaria Adamo, Caterina Lo Rizzo, Mirella Bruttini, Alessandra Renieri, and Francesca Mari

Corresponding author: Francesca Mari (francesca.mari@unisi.it)

Review Timeline:

Submission Date:	17th Aug 23
Editorial Decision:	19th Sep 23
Revision Received:	29th Dec 23
Editorial Decision:	22nd Jan 24
Revision Received:	10th Feb 24
Accepted:	12th Feb 24

Editor: Zeljko Durdevic

Transaction Report:

19th Sep 2023

Dear Prof. Mari,

Thank you for the submission of your manuscript to EMBO Molecular Medicine. We have now received feedback from a reviewer who agreed to evaluate your manuscript. The referee recognizes potential interest of the study but also raises a number of concerns that should be addressed in a major revision. If you would like to discuss further the points raised by the referees, I am available to do so via email or video. Let me know if you are interested in this option.

Further consideration of a revision that addresses reviewer's concerns in full will entail a second round of review. EMBO Molecular Medicine encourages a single round of revision only and therefore, acceptance or rejection of the manuscript will depend on the completeness of your responses included in the next, final version of the manuscript. For this reason, and to save you from any frustrations in the end, I would strongly advise against returning an incomplete revision.

We would welcome the submission of a revised version within three months for further consideration. Please let us know if you require longer to complete the revision.

I look forward to receiving your revised manuscript.

Yours sincerely,

Zeljko Durdevic

We require:

- 1) A .docx formatted version of the manuscript text (including legends for main figures, EV figures and tables). Please make sure that the changes are highlighted to be clearly visible.
- 2) Individual production quality figure files as .eps, .tif, .jpg (one file per figure). For guidance, download the 'Figure Guide PDF': (<https://www.embopress.org/page/journal/17574684/authorguide#figureformat>).
- 3) A .docx formatted letter INCLUDING the reviewers' reports and your detailed point-by-point responses to their comments. As part of the EMBO Press transparent editorial process, the point-by-point response is part of the Review Process File (RPF), which will be published alongside your paper.
- 4) A complete author checklist, which you can download from our author guidelines (<https://www.embopress.org/page/journal/17574684/authorguide#submissionofrevisions>). Please insert information in the checklist that is also reflected in the manuscript. The completed author checklist will also be part of the RPF.
- 5) Please note that all corresponding authors are required to supply an ORCID ID for their name upon submission of a revised manuscript.
- 6) It is mandatory to include a 'Data Availability' section after the Materials and Methods. Before submitting your revision, primary datasets produced in this study need to be deposited in an appropriate public database, and the accession numbers and

database listed under 'Data Availability'. Please remember to provide a reviewer password if the datasets are not yet public (see <https://www.embopress.org/page/journal/17574684/authorguide#dataavailability>).

13) Author contributions: You will be asked to provide CRediT (Contributor Role Taxonomy) terms in the submission system. These replace a narrative author contribution section in the manuscript.

14) A Conflict of Interest statement should be provided in the main text.

15) Every published paper now includes a 'Synopsis' to further enhance discoverability. Synopses are displayed on the journal

webpage and are freely accessible to all readers. They include a short stand first (maximum of 300 characters, including space) as well as 2-5 one-sentences bullet points that summarizes the paper. Please write the bullet points to summarize the key NEW findings. They should be designed to be complementary to the abstract - i.e. not repeat the same text. We encourage inclusion of key acronyms and quantitative information (maximum of 30 words / bullet point). Please use the passive voice. Please attach these in a separate file or send them by email, we will incorporate them accordingly.

Please note: When submitting your revision you will be prompted to enter your funding and payment information. This will allow Wiley to send you a quote for the article processing charge (APC) in case of acceptance. This quote takes into account any reduction or fee waivers that you may be eligible for. Authors do not need to pay any fees before their manuscript is accepted and transferred to the publisher.

EMBO Press participates in many Publish and Read agreements that allow authors to publish Open Access with reduced/no publication charges. Check your eligibility: <https://authorservices.wiley.com/author-resources/Journal-Authors/open-access/affiliation-policies-payments/index.html>

***** Reviewer's comments *****

Referee #1 (Remarks for Author):

The short report by Maffeo et al, on a new two generation family (mother and son) with a hitherto not reported heterozygous MET mutation c.3704A>G/ p.Tyr1235Cys expands the clinical features of the previous report by Zhou et al, 2019 on a first family with distal arthrogryposis involving only the upper limbs and associated with MET p.Tyr1234Cys mutation. This adds to our knowledge about the underlying genetic causes of arthrogryposis.

I nevertheless have two major comments.

The clinical description is very limited. Joint contractures are the consequence of decreased fetal movement and there is a wide range of limb positions even within families. Joint contractures should therefore be explicitly described (flexion/ extension contractures of interphalangeal joints, hallux valgus deformity, etc.). The position of the thumb is unusual for classic distal arthrogryposis and the authors should state if there is hyperextension of the interphalangeal joint of the thumbs or clinodactyly and if there is congenital limitation of joint range of motion - it does not look very pathologic on the photographs. It would also be interesting to know if there were other features of fetal hypokinesia (pterygia, dimples...). Prenatal ultrasonography findings are rare in arthrogryposis multiplex congenita and especially so in distal arthrogryposis. Please provide if possible the precise findings of the prenatal ultrasound. It would also be of interest to know if the findings on the lower limbs of the mother have appeared over time, or were present at birth. Please also mention that you have received explicit consent from the family to publish the photographs. The article by Zhou et al, 2019 provides axial T1 forearm muscle MRIs that might help guide the clinical diagnosis. Please mention at least if MRIs have been performed or not.

Please provide more information on the exome sequencing data: potential other candidate genes that were found, family segregation (who was sequenced - the boy, the mother and boy, both parents and the boy, other family members?), quality control.

**** Reviewer's comments ****

Referee #1 (Remarks for Author):

The short report by Maffeo et al, on a new two generation family (mother and son) with a hitherto not reported heterozygous MET mutation c.3704A>G/ p.Tyr1235Cys expands the clinical features of the previous report by Zhou et al, 2019 on a first family with distal arthrogryposis involving only the upper limbs and associated with MET p.Tyr1234Cys mutation.

This adds to our knowledge about the underlying genetic causes of arthrogryposis.

I nevertheless have two major comments.

The clinical description is very limited.

R: We have added several details of the clinical picture of the two patients following your indication.

Joint contractures are the consequence of decreased fetal movement and there is a wide range of limb positions even within families. Joint contractures should therefore be explicitly described (flexion/ extension contractures of interphalangeal joints, hallux valgus deformity, etc.).

R: we have added the requested information as follows "The proband, an 18-year-old boy, presented bilateral hand arthrogryposis, diagnosed during pregnancy. He presented with diffuse arthrogryposis in both hands with inability to completely extend fingers, absent flexion creases bilaterally, bilateral thenar and hypothenar hypotrophy and bilateral muscular hypotrophy of the distal two thirds of the forearm, functional limitation of the elbow joint (more visible on the left side) and limited forearm supination. On the lower limb he presented bilateral pes cavum and hallux valgus. thenar, hypothenar and forearm muscles hypotrophy and clinodactyly of the distal phalanx of the first ray of both hands (Fig 1B, IV:3). Specifically, in the right hand, he presented functional limitation of the proximal interphalangeal joint of the first ray and limited flexion of the distal interphalangeal joint of the first ray, clinodactyly and flexion limitation of the interphalangeal joints of the second ray, flexion limitation of all interphalangeal joints of third and fourth ray, mild clinodactyly of the fifth ray with extension and flexion limitation. In the left hand he presented functional limitation of the proximal interphalangeal joint of the first ray, clinodactyly of the second ray, partially reducible, functional limitation of distal interphalangeal joints of the third and fourth ray, clinodactyly of the fifth ray. He presented a scar on the ulnar side due to a previous orthopedic surgery. His mother presented a more severe form, also involving feet, with overriding fingers (Fig 1B, III:4). She presented flexion limitation of all digits of both hands, bilateral thenar and hypothenar hypotrophy and distal forearm muscular hypotrophy (more severe on the right side). In the right hand, the most severely affected, she reported a previous orthopedic surgery between the first and second ray to improve grip, she presented camptodactyly of the second ray due to a pterygium, she reported a previous orthopedic surgery of the V ray to possibly remove a pterygium that was totally limiting digit extension. On the feet, she showed the second finger overlapping the third bilaterally already present at birth, bilateral pes cavum. She reported a previous surgery for a third mallet toe on the left foot. MRIs have not been performed in any of the two affected individuals."

The position of the thumb is unusual for classic distal arthrogryposis and the authors should state if there is hyperextension of the interphalangeal joint of the thumbs or clinodactyly and if there is congenital limitation of joint range of motion - it does not look very pathologic on the photographs.

R: the thumb is in hyperextension that can be easily reduced. We added new images that confirm this statement. There is indeed flexion limitation.

It would also be interesting to know if there were other features of fetal hypokinesia (pterygia, dimples...).

R: There were pterygia in the mother in the II and V ray of the right hand.

Prenatal ultrasonography findings are rare in arthrogryposis multiplex congenita and especially so in distal arthrogryposis. Please provide if possible the precise findings of the prenatal ultrasound.

R: mother reported that the gynecologist when the baby was born told her that in the last fetal ultrasound, he saw signs of arthrogryposis. No medical reports state that. We have eliminated the sentence "diagnosed during pregnancy" in the manuscript.

It would also be of interest to know if the findings on the lower limbs of the mother have appeared over time, or were present at birth.

R: The findings of the lower limbs of the mother were present at birth. Following your concern and the fact that the feet abnormalities are not completely consistent in the two affected individuals, in the manuscript we have added a sentence arguing if the feet abnormalities could be linked to something else and not be part of the MET-related phenotype.

Please also mention that you have received explicit consent from the family to publish the photographs.

R: Explicit consent for photographs publications has been received by all subjects. This is now stated in the manuscript. The article by Zhou et al, 2019 provides axial T1 forearm muscle MRIs that might help guide the clinical diagnosis.

Please mention at least if MRIs have been performed or not.

R: MRIs have not been performed in any of the two affected individuals.

Please provide more information on the exome sequencing data: potential other candidate genes that were found, family segregation (who was sequenced - the boy, the mother and boy, both parents and the boy, other family members?), quality control.

R: Additional information has been added in the manuscript. No additional potential candidate genes have been identified. Clinical exome has been performed in the two affected subjects (proband and his mother). Sanger sequencing confirmed the mutation in the two affected subjects and in none of the tested healthy family members (brother and maternal grandparents), indicating the possible *de novo* occurrence of the mutation in the proband's mother and confirming the pathogenetic role of the identified variant.

22nd Jan 2024

Dear Prof. Mari,

Thank you for the submission of your manuscript to EMBO Molecular Medicine. I am pleased to inform you that we will be able to accept your manuscript pending the following final amendments:

- 1) Please address all referee's points. Adding additional photographs of the upper limb (point #2) and MRI images (point #5) is welcomed but not essential for publication of the manuscript.
- 2) In the main manuscript file, please do the following:
 - Add callouts for Figure 1A and 1D.
 - Please rename "Conflict of interest" to "Disclosure Statement & Competing Interests". We updated our journal's competing interests policy in January 2022 and request authors to consider both actual and perceived competing interests. Please review the policy <https://www.embopress.org/competing-interests> and update your competing interests if necessary.
 - Author contributions: Please remove it from the manuscript and specify author contributions in our submission system. CRediT has replaced the traditional author contributions section because it offers a systematic machine-readable author contributions format that allows for more effective research assessment. You are encouraged to use the free text boxes beneath each contributing author's name to add specific details on the author's contribution. More information is available in our guide to authors: <https://www.embopress.org/page/journal/17574684/authorguide#authorshipguidelines>
- 3) As part of the EMBO Publications transparent editorial process initiative (see our Editorial at <http://embomolmed.embopress.org/content/2/9/329>), EMBO Molecular Medicine will publish online a Review Process File (RPF) to accompany accepted manuscripts. This file will be published in conjunction with your paper and will include the anonymous referee reports, your point-by-point response and all pertinent correspondence relating to the manuscript. Let us know whether you agree with the publication of the RPF and as here, if you want to remove or not any figures from it prior to publication. Please note that the Authors checklist will be published at the end of the RPF.
- 4) Please provide a point-by-point letter INCLUDING my comments as well as the reviewer's reports and your detailed responses (as Word file).

I look forward to reading a new revised version of your manuscript as soon as possible.

Yours sincerely,

Zeljko Durdevic

*** Instructions to submit your revised manuscript ***

To submit your manuscript, please follow this link:

<https://embomolmed.msubmit.net/cgi-bin/main.plex>

- 1) a .docx formatted version of the manuscript text (including Figure legends and tables)

2) Separate figure files*

3) supplemental information as Expanded View and/or Appendix. Please carefully check the authors guidelines for formatting Expanded view and Appendix figures and tables at <https://www.embopress.org/page/journal/17574684/authorguide#expandedview>

4) a letter INCLUDING the reviewer's reports and your detailed responses to their comments (as Word file).

5) The paper explained: EMBO Molecular Medicine articles are accompanied by a summary of the articles to emphasize the major findings in the paper and their medical implications for the non-specialist reader. Please provide a draft summary of your article highlighting

6) For more information: There is space at the end of each article to list relevant web links for further consultation by our readers. Could you identify some relevant ones and provide such information as well? Some examples are patient associations, relevant databases, OMIM/proteins/genes links, author's websites, etc...

7) Author contributions: the contribution of every author must be detailed in a separate section.

8) EMBO Molecular Medicine now requires a complete author checklist (<https://www.embopress.org/page/journal/17574684/authorguide>) to be submitted with all revised manuscripts. Please use the checklist as guideline for the sort of information we need WITHIN the manuscript. The checklist should only be filled with page numbers where the information can be found. This is particularly important for animal reporting, antibody dilutions (missing) and exact values and n that should be indicated instead of a range.

9) Every published paper now includes a 'Synopsis' to further enhance discoverability. Synopses are displayed on the journal webpage and are freely accessible to all readers. They include a short stand first (maximum of 300 characters, including space) as well as 2-5 one sentence bullet points that summarise the paper. Please write the bullet points to summarise the key NEW findings. They should be designed to be complementary to the abstract - i.e. not repeat the same text. We encourage inclusion of key acronyms and quantitative information (maximum of 30 words / bullet point). Please use the passive voice. Please attach these in a separate file or send them by email, we will incorporate them accordingly.

You are also welcome to suggest a striking image or visual abstract to illustrate your article. If you do please provide a jpeg file 550 px-wide x 300-800px high.

10) A Conflict of Interest statement should be provided in the main text

11) Please note that we now mandate that all corresponding authors list an ORCID digital identifier. This takes <90 seconds to complete. We encourage all authors to supply an ORCID identifier, which will be linked to their name for unambiguous name identification.

Currently, our records indicate that the ORCID for your account is 0000-0003-1992-1654.

Link Not Available

Photos 400-800 DPI

*Additional important information regarding figures and illustrations can be found at <https://bit.ly/EMBOPressFigurePreparationGuideline>. See also figure legend preparation guidelines: <https://www.embopress.org/page/journal/17574684/authorguide#figureformat>

**** Reviewer's comments ****

Referee #1 (Remarks for Author):

Maffeo et al, have added substantial clinical data and photographs to improve the manuscript.

I have nevertheless some minor remarks:

1/ Please consider ACMG guidelines when reporting molecular data.

2/ Please extend photographs to the upper limb including elbow and forearm in order to show amyotrophy of the distal part of the forearm in addition to the joint contractures of the hand. This would add additional data compared to Zhou et al, 2019.

3/ Consider shortening and please correct (there is no proximal and distal interphalangeal joint of the first ray) the two sentences "Specifically, in the right hand... In the left hand he presented...".

4/ Please replace "functional limitation" used at several occasions (elbow, thumb) by a precise statement of joint limitation to be able to understand what is limited at rest (flexion contracture of an interphalangeal joint would mean a position at rest with an irreducible bent towards the palm of the hand, and could also be described as a limited extension of the interphalangeal joint). You could easily use HPO terms.

5/ Please consider at least in one of the two patients a muscle MRI of the forearm and hand. This should not be too difficult to obtain and would further improve the manuscript and the comparison with the article by Zhou et al, 2019. A whole body MRI, instead of a MRI of the forearm and hand, could also search for selective muscle atrophies in the lower limbs, especially in the leg.

Reviewer's comments and authors' response

Referee #1 (Remarks for Author):

Maffeo et al, have added substantial clinical data and photographs to improve the manuscript.

I have nevertheless some minor remarks:

1/ Please consider ACMG guidelines when reporting molecular data.

R: the genetic variant has been reported using the standard gene variant nomenclature maintained at the Human Genome Variation Society (HGVS) as indicated by the ACMG.

2/ Please extend photographs to the upper limb including elbow and forearm in order to show amyotrophy of the distal part of the forearm in addition to the joint contractures of the hand. This would add additional data compared to Zhou et al, 2019.

R: we were able to add additional pictures of the upper limbs showing amyotrophy of the distal part of the forearm.

3/ Consider shortening and please correct (there is no proximal and distal interphalangeal joint of the first ray) the two sentences "Specifically, in the right hand... In the left hand he presented...".

R: we apologize for the oversight. We have shortened and corrected the indicated sentences.

4/ Please replace "functional limitation" used at several occasions (elbow, thumb) by a precise statement of joint limitation to be able to understand what is limited at rest (flexion contracture of an interphalangeal joint would mean a position at rest with an irreducible bent towards the palm of the hand, and could also be described as a limited extension of the interphalangeal joint). You could easily use HPO terms.

R: we have better specified the joint limitation.

5/ Please consider at least in one of the two patients a muscle MRI of the forearm and hand. This should not be too difficult to obtain and would further improve the manuscript and the comparison with the article by Zhou et al, 2019. A whole body MRI, instead of a MRI of the forearm and hand, could also search for selective muscle atrophies in the lower limbs, especially in the leg.

R: we have been able to add the pictures of the forearm of proband and his mother. We were not able to add MRI images since they are not available.

12th Feb 2024

Dear Prof. Mari,

We are pleased to inform you that your manuscript is accepted for publication and is now being sent to our publisher to be included in the next available issue of EMBO Molecular Medicine.
